# Laboratory Compaction Study and Mechanical Performance Assessment of Half-Warm Mix Recycled Asphalt Mixtures Containing 100% RAP

**DOI:** 10.3390/ma12121992

**Published:** 2019-06-21

**Authors:** José Marcobal, José Lizárraga, Juan Gallego

**Affiliations:** 1Department of Pavements, Sacyr Construction, Pº de la Castellana, 83-85, 28046 Madrid, Spain; jrmarcobal@sacyr.com; 2Department of Transport Engineering, Urban and Regional Planning, Technical University of Madrid (UPM), C/Profesor Aranguren s/n, 28040 Madrid, Spain; juan.gallego@upm.es

**Keywords:** half-warm mix recycled asphalt mixtures, 100% recycling, curing treatment, stiffness modulus, rutting, fatigue

## Abstract

The use of low-carbon and energy-efficient paving technologies is gaining worldwide acceptance in recent years as a means to encourage commitment towards more sustainable pavement management practices. However, there still remain some technical gaps regarding mix design procedures for the half-warm mix asphalt (HWMA) mixtures’ preparation and characterization in the laboratory. To this end, three different laboratory compaction methods (e.g., static load, Marshall impactor, and gyratory compactor) were selected and put into assessment to define the most suitable compaction test method for half-warm mix recycled asphalt (HWMRA) mixtures with 100% reclaimed asphalt pavement (RAP). Posteriorly, the effect of four-accelerated curing treatments (0, 24, 48, and 72 h) on the mixtures’ mechanical performance was investigated. Then, advanced mechanical characterization of the mixture performance was conducted to quantify the indirect tensile strength (ITS), stiffness modulus, rutting, and four-point bending (4PB) fatigue test. Thus, based on the authors’ findings, the HWMRA mixtures with 100% RAP and emulsified bitumen exhibited proper volumetric (e.g., air voids and density) and mechanical behavior in terms of moisture damage, ITS, stiffness modulus, rutting, and fatigue cracking. These findings encourage greater confidence in promoting the use of these sustainable asphalt mixes for their use in road pavements or urban streets.

## Highlights 

Half-warm mix asphalt (HWMA) mixtures with 100% RAP and emulsified bitumen were designedA mix design compaction effort of 70 gyros was selected for simulating field compaction conditionsA curing treatment of 72 h, at 50 °C, was used for half-warm mix’ optimization and characterizationThe optimum emulsion content (OEC) was defined as a function of the mixtures’ volumetric propertiesThe HWMRA mixtures showed adequate mechanical performance properties

## 1. Introduction

Over the last few decades, the transition from a high-consuming carbon and linear society to a closed-loop circular economy (CE) is gaining worldwide boost as a way to move towards more sustainable pavement management practices by positioning the use of greener disruptive production technologies at the top level of the agenda for sustainable development [1]. One example of a technology that has the potential to decrease the consumption of energy sources (i.e., fuel and gas-oil) and the extraction of raw materials is the use of half-warm mix asphalt (HWMA) mixture with recycled asphalt pavement (HWMRA). In fact, HWMA mixes are manufactured, spread, and compacted in the working temperature range of 60–100 °C, whereas conventional hot mix asphalt (HMA) mixes are manufactured at 160–170 °C, warm mix asphalt (WMA) in the temperature range of 100–140 °C, and cold mix asphalt (CMA) below 60 °C [2,3]. HWMA mixes offer many other potential benefits, such as lower energy consumption of up to 3–4 kg/ton—compared with both hot in-place recycling (HIR) and warm mix asphalt (WMA) [4] and decrease in carbon footprint emissions during mix production, by 30% of CO2 and 25% of SO2, in comparison with conventional mixtures [5]. 

Despite the environmental and economic advantages offered by this technology, a large number of questions remain regarding their compactibility, namely since mix compaction occurs above the recycled binder’s softening point temperature (~ 85 °C), the RAP aggregates tend to behave as black rock (i.e., there is no blending between aged RAP binder and emulsion), implying a higher plastic deformation behavior of the recycled aggregates compared to the solid–rigid behavior expected from virgin aggregates used either in cold in-place (CIR) or hot in-place recycling (HIR) [6]; whilst some other authors reported that mixture compactibility improves as a result of the combination of half-warm temperatures and water content provided by the emulsion, which makes it possible to achieve a better quality of mixture in the field [7].

On the other hand, recent investigations have shown that reducing manufacturing temperatures within the range of warm and half-warm temperatures and incorporating high rates of RAP might not affect nor compromise the mechanical performance properties of the mixtures [8,9]. Despite this, some other authors claim that reducing production and compaction temperatures leads to inadequate volumetric characteristics, compaction deficiency, and moisture sensitivity, likely due to the weakening of the adhesive interface bonding between aggregates and binder [10,11]. Nevertheless, some authors stand up for the idea that, as the RAP aggregates are already covered by a thin-film layer of aged RAP binder, the moisture susceptibility is not expected to be worse than conventional HMA mixtures [12,13]; whilst some other authors asserted that mixtures with high RAP contents and at low temperatures showed lower load-bearing capacity and rutting performance values, likely as a result of lowering mixing and compaction temperatures [14,15]. 

Additionally, some works claim that the reuse of high or total recycled binder contents tends to make the mixture stiffer and more brittle because of the physical hardening and oxidative aging suffered by the recycled RAP binder during its service life. In other words, the RAP mixtures will exhibit lower indirect tensile strength (ITS) values (i.e., due to the formation of a heterogeneous binder that prevents the full-blending process between recycled aggregates and virgin binder) [16], increased stiffness modulus, less resistance to fatigue cracking due to chemical aging and hardening process [17], low-temperature brittleness [18], decreased ductility [19], loss of uniformity of the recycled aggregates and gradation [20]; and higher the RAP content added in the mix design resulting in lower tensile fatigue–strain deformation [21]. 

Nonetheless, and contrary to popular beliefs, the study of the issues related to the fatigue cracking resistance of mixtures with RAP has led to contradictory results. For instance, some authors reported that lowering the production temperatures allowed to decrease the RAP aging process of the innermost thin-asphalt layer adhered to the recycled aggregates, thereby obtaining a less stiff and less brittle binder, while avoiding the decrease in its resistance to fatigue cracking [22,23]. To support this, some researchers have shown that mixes with 100% RAP at half-warm temperatures can exhibit similar resistance to fatigue cracking, stiffness, and ductility at three testing temperatures (20 °C, 5 °C, and −5 °C), in comparison with the values expected from conventional HMA mixtures [24], regardless of the failure criterion used for the mechanical analysis (i.e., semi-circular bending (SCB) fracture test, uniaxial tension-compression strain sweep, and indirect tensile fatigue test) [25,26,27,28]. In this line, Lopes et al. [29] reported that the addition of RAP improves the fatigue life of such mixtures since the recycled asphalt binder forms a stiff microlayer at the interface of RAP that reduces both the strain and stress concentration within the mixture [30]. 

However, high RAP content mixtures are still downgraded in base and binder course asphalt mixtures, due to the stringent requirements stipulated for this layer in terms of rutting and skid resistance [31]. For instance, the Spanish Ministry of Public Works, through circular order (OC 40/2017: Recycling of Pavements), restricts the maximum allowable RAP content for hot in-plant recycling (HIR) of bituminous mixtures for its use in binder and wearing course asphalt mixtures and low traffic categories [32]. The likely explanation lies in the constant uncertainty regarding long-term mechanical performance, a good understanding of the blending and interaction process between aged RAP binder and virgin binder [33,34], variability of RAP properties obtained from different sources [35], and that the use of RAP may lead to a lower durability, increased stiffness, and early fatigue cracking pavement failures. 

In this context, and due to contradictory findings, a more in-depth laboratory compaction study is, therefore, necessary to evaluate the volumetric characteristics, compactibility and mechanical performance properties of these mixtures. In order to do this, a laboratory research project was launched by using public financial support to make up for current Spanish regulatory gaps and, in turn, put into assessment a new cleaner production technology based on the revalorization and reuse of recycled asphalt pavements; where by the main contributions of this study were found to be the adoption of the gyratory compactor system as the most suitable method for this technology, and the use of an accelerated curing treatment that allowed to improve the mixtures’ mechanical performance properties in the curing range of48-72 h, at 50 °C. 

## 2. Objectives

The objectives of this research paper have been twofold. The first objective was to define the mix design compaction effort (N*design*) that allows to simulate and reproduce the laboratory volumetric characteristics and mechanical performance properties in the field. Having detected this gap, the second aim was to characterize and present the results from the effect of laboratory-accelerated curing treatment on the development of the mechanical performance (e.g., indirect tensile strength, stiffness modulus, rutting, and fatigue cracking) of half-warm mix recycled asphalt (HWMRA) mixtures with a total RAP content of 100% and emulsified bitumen.

## 3. Methodology

This research study has been broken down into four phases. In the first phase, a preliminary laboratory study was conducted to characterize the RAP that was used in the production of half-warm mix recycled asphalt (HWMRA) mixtures with emulsified bitumen. The second phase consisted of evaluating and comparing three different compaction test methods (i.e., Marshall impactor, static compressive load, and gyratory compactor) to define the most suitable compaction test method and hence the mix design compaction effort that allows to match the specimens’ benchmark density with the density expected from the field cores. Additionally, the mix design procedure was based on the comparison between the (1) immersion–compression (I-C) test (NLT 162/00: Effect of Water on Compressive Strength of Compacted Bituminous Mixtures), and (2) the indirect tensile strength ratio (ITSR), according to EN 12697-12:2018. Part 12: Determination of the water sensitivity of bituminous specimens. 

In the third stage, a preliminary research study was conducted to quantify the influence of five emulsion contents (0%, 2.0%, 2.5%, 3.0%, and 3.5%o/RAP) on the volumetric and mechanical performance properties of the mixtures in terms of bulk density, by saturated surface dry (SSD) conditions, air voids, stiffness modulus, and indirect tensile strength. For this purpose, an average of three cylindrical specimens for each emulsion content was prepared (with a diameter of 100 mm and 63 mm in height) and tested to determine the optimal emulsion content (OEC) that allows better optimization of the ultimate mix design performance. Thus, an assessment of the effect of four laboratory-accelerated curing/drying heating treatments (0, 24, 48, and 72 h) on the half-warm mixes was conducted to determine, as part of the optimization process, the possible improvement of the mechanical performance properties (e.g., indirect tensile strength and stiffness modulus) of the mixtures. In the fourth phase, an advanced mechanical characterization of mixtures was conducted based on four different behavior criteria, such as (1) stiffness modulus, (2) indirect tensile strength (ITS), (3) rutting performance with the wheel tracker, and (4) fatigue cracking assessed via four-point (4PB) bending beam test method. [36]. Figure 1 illustrates the main four phases/stages of the experimental research methodology addressed in this research study.

## 4. Test Procedures

Initially, the binder extraction and recovery tests were conducted on both coarse- and fine-aggregate RAP fractions to obtain the corresponding content of recycled asphalt binder, according to EN 12697-1:2012. Part 1: Soluble binder content, and EN 12697-3:2013. Part 3: Bitumen recovery–Rotary evaporator. 

The mixtures’ volumetric characteristics were determined in the laboratory, according to EN 12697-8:2003. Part 8: Determination of void characteristics of bituminous specimens using the bulk density, by saturated surface dry (SSD) conditions, according to EN 12697-6:2012. Part 6: Determination of bulk density of bituminous specimens, and the maximum density was obtained using a pycnometer, according to EN 12697-5:2010. Part 5: Determination of the maximum density - Procedure A: Volumetric method. An average of three cylindrical shaped specimens was prepared and manufactured to determine the bulk density while the maximum density was calculated using two asphalt samples with the pycnometer. 

The compaction test methods that were selected for specimen’s compaction and characterization in the laboratory were (1) the vertical static load by double-plunger, according to NLT 161/98: Standard Test Method for Compressive Strength of Compacted Bituminous Mixtures; (2) Marshall impactor hammer, according to EN 12697-30:2012. Part 30. Specimen preparation by impact compactor; and (3) the Gyratory Compactor, according to EN 12697-31:2007. Part 31: Specimen preparation by Gyratory Compactor. 

Additionally, the laboratory tests selected to evaluate the resistance to water action and mechanical performance properties of the mixtures studied were as follows: (a) the determination of the water sensitivity of bituminous specimens, according to EN 12697-12:2009; and (b) the immersion–compression (I-C) test, according to NLT-162/00. For the I-C test, an average of eight cylindrical shaped laboratory specimens was prepared with a diameter of 101.6 mm and a height of 100 mm and compacted with a static contact load pressure produced by double-plunger action. The initial pre-loading applied was approximately 1 MPa, and, hence, the load starts gradually increasing until reaching 20.7 MPa (3000 psi), maintaining the vertical contact load pressure for 2 min. The first four specimens are left at room temperatures (25 °C) for 24 h. The other subset was immersed in a water bath for 24 h, at 60 °C. Posteriorly, the specimens were placed in a water bath, at 25 °C, for 2 h. Both subsets were subjected to a simple compressive load at a constant deformation rate of 5.08 mm/min. Afterward, the percentage of retained water strength resistance is calculated between the wet and dry subsets, which makes it possible to obtain the resistance to moisture damage of the specimens. The optimal percentage should meet the minimum retained water strength requirements, depending on the levels of heavy traffic load to be supported. 

On the other hand, the water sensitivity test consisted in manufacturing a set of six cylindrical specimens, with a diameter of 101.6 mm and 63.5 mm in height, and compacted by the gyratory compactor using two-thirds of the benchmark compaction energy previously selected, and following the standard compaction conditions (0.82°, 600 kPa, and 30 rpm) established by EN 12697-31:2008. Part 31: Gyratory Compactor. The specimens were classified into two subsets: (1) a dry subset stored, at 20 °C, for 72 h, and the wet subset immersed in-water bath conditions, at 40 °C, during the same period after a vacuum pressure of 6.7 ± 0.3 MPa. Following the laboratory standard, the indirect tensile strength (ITS) test was carried out (EN 12697-12:2018. Part 12: Water sensitivity) to calculate the indirect tensile strength ratio (%, ITSR) between the wet and dry subset (EN 12697-23:2018.Part 23: Determination of the indirect tensile strength of bituminous specimens); whereby the ITS test consisted in subjecting the specimens to diametral compressive strength loads using two loading strips (with a width of 12.7 mm) at a constant deformation rate of 50 ± 2 mm/min, at 15 °C, in which this load produced a tensile stress load through the vertical diametral plane. 

Concerning the advanced mechanical characterization of the mixtures studied, the rutting performance, stiffness modulus, and fatigue cracking resistance by four-point (4PB) flexural bending beam test method were evaluated in the laboratory. The resistance to permanent deformation was assessed by conducting the wheel-tracking test (WTT), according to EN 12697-22:2008+A1:2008. Part 22: Wheel Tracking. The rutting test consisted in applying a total of 10.000 load cycles, at a frequency of 26.5 ± 1 load cycles/minute, procedure B, in air, using a loaded rubber wheel back and forth on the prismatic specimen with a contact load of 700 N. For this test, an average of two prismatic-shaped specimens were prepared (with a length of 400 mm, 250 in width, and 60 mm in height) and compacted with a percentage of 98% of the benchmark density using the steel roller compactor device, according to EN 12697-33:2006. Part 33: Specimen prepared by roller compactor.

The mixtures’ load-bearing capacity was assessed through the indirect tensile stiffness modulus (Sm), at 20 °C, according to EN-12697-26:2012. Part 26: Stiffness. This property was calculated as the average stiffness value of five indirect-tensile haversine-shaped load waveform pulses on a diametral section with a rise time of 124 ± 3 ms, peak load adjusted until horizontal deformation of 5µm, loading frequency of 2.1 Hz, peak loading force of 1000 N, and Poisson’s ratio (ν) of 0.35; where 10 load pulses are previously applied to set up the system in terms of loading level and frequency. Therefore, the average stiffness modulus value was validated and contrasted by turning the cylindrical specimen at an angle of 90 ± 10°, according to its longitudinal axis on the plate. Thus, for an applied dynamic load of P in which the resulting horizontal dynamic deformations are determined, the total stiffness modulus is calculated using Equation (1) [37]
(1)$$S_{m}=\frac{P\left(\gamma +0.27\right)}{t\delta_{h}}$$
where: Sm: stiffness modulus, MPa; P: maximum dynamic load, N; γ: Poisson’s ratio (0.35); t: specimen thickness, mm; $\delta_{h}$: total horizontal recoverable deformation expressed in terms of mm.

The mixtures’ fatigue cracking analysis was conducted using the four-point flexural bending (4PB) beam fatigue test, at 20 °C, according to EN 12697-24:2013. Part 24: Resistance to fatigue - Annex D. To do so, more than 12 laboratory prismatic-shaped specimens for each type of asphalt mixture were compacted with the steel roller compactor and thereafter sawed (with a length of 380 mm, 50 mm in width and 50 mm in height), for their posterior testing in a 4PB device. Following the production/compaction process, the 4PB fatigue strength test was conducted applying harvesine-shaped load pulses in strain–fatigue control mode and with a selected loading frequency of 30 Hz. In turn, the vertical deflection at the center of the beam was measured using a linear variable differential transducer (LVDT) positioned at the bottom of the prismatic specimen. The controlled fatigue–strain amplitude levels selected for the 4PB test varied from 200–250 µm/m, 150–190 µm/m, and 100–140 µm/m; where the fatigue life should fall within the range of 10^4^ and 2 × 10^6^ load cycles. Thus, the two parameters selected to depict the mixtures’ fatigue cracking resistance were the number of load cycles, N*f*, to failure and the corresponding tensile fatigue–strain level ($\varepsilon_{t}$). Moreover, the fatigue failure approach was defined using the classical fatigue method expressed by a relationship between the tensile strain ($\varepsilon_{t}$) and the number of load cycles to failure, N*f*, at which the initial stiffness modulus of the specimens measured in the load cycle number 100 is reduced to 50% of its initial beam stiffness [38,39,40]. This procedure was based on Equation (2)
(2)$$\varepsilon_{t}=A\cdot {\left(N_{f}\right)}^{B}$$
where $\varepsilon_{t}$ is the tensile strain, με, applied in the center of the prismatic specimens; N*f* is the number of load cycles to failure; A and B are the coefficients of the material determined in the laboratory based on the type of material.

## 5. Materials

### 5.1. RAP Characterization

The reclaimed asphalt pavement (RAP) material was recovered from a test road section and classified in two fractions: coarse (5/25 mm) and fine (0/5 mm). Ts, both RAP fractions were homogenized, quartered, treated, and characterized to determine their residual binder content, through the centrifuge extractor method (EN 12697-1:2012). For the coarse fraction 5/25 mm (60%) the residual binder content was found to be 2.60 (%, o/RAP), whereas, for the fine fraction 0/5 mm (40%), this content was 6.45 ± 0.1 (%, o/over the weight of RAP). As a result, the content of the aged binder in the RAP (2.60 * 0.6 + 6.45 * 0.4) was 4.14% over the weight of RAP. Following the dosing procedure, 2.5% o/RAP of emulsion (with 60% residual asphalt binder) was added to the RAP material (60% * 2.5%), 1.5% of residual asphalt binder is obtained and added in the aged RAP binder (4.14%o/RAP), resulting in a total residual binder content of 5.64% for 2.5% emulsion; whilst, for 3.0% emulsion, this content was found to be 5.94%(o/RAP). 

Once the residual binder content was determined, the next step was to analyze the binder’ consistency properties in terms of penetration test (EN 1426:2015) and softening point temperature, by ring and ball (R&B) method, according to EN 1427:2015; where the average penetration value of the aged RAP binder was 17 dmm and softening point temperature of 67.2 °C. Moreover, white and black RAP aggregate grading curves were also determined; namely, the white curves can be defined as the RAP gradation after extracting the residual aged binder while the black curves represent the RAP gradation containing the recycled asphalt binder. Figure 2 depicts the black and white RAP grading curves of both recycled aggregate fractions (0/5 and 5/25 mm) in which the dashed lines represent the black grading curves, while the solid continuous lines depict the white grading curves.

### 5.2. Bituminous Emulsion Characterization

In this research study, two cationic slow-setting bitumen emulsions (C60B5) were formulated and selected, depending on the penetration grade bitumen to be used, i.e., (1) a conventional 50/70 pen. grade with a residue bitumen content of 61.2% by total weight of the emulsion; and (2) a second emulsion made up of a rejuvenator binder of 160/220 penetration grade bitumen; where the bituminous emulsion meets the current specifications of the framework for specifying cationic bituminous emulsions, according to EN 13808:2013. The characterization of cationic emulsion consisted of the analysis of the viscosity, at 25 °C, water content from the emulsion, residue on sieving, residual binder content, and penetration test of the residual binder. Table 1 shows the general technical characteristics of the cationic bituminous emulsion used to produce both HWMRA mixtures.

### 5.3. Aggregate Grading Curve

The particle size distribution of the recycled material fell within the threshold sieve size values stipulated by the Art. 20 of PG-4: In-situ recycling of asphalt mixtures with bitumen emulsion [32]. The proportion of RAP aggregates, after the screening, was determined to be 40% in the fine fraction (0/5 mm) and 60% in the coarse fraction (5/25mm). This proportion was selected (1) in order to deal with a RAP content equal to 100%; (2) to ensure the mixture’ homogeneity (i.e., control mixture quality, fines particles and mastic content in the mixture design); and (3) to meet the RE2 aggregate gradation band, according to Art. 20 of PG-4: “In-situ recycling of bituminous layers with emulsion” [32]. Table 2 and Figure 3 show the selected recycled aggregate grading curve of the HWMRA 100% RAP mixture, as well as the upper and lower threshold values of the RE2 particle size distribution band, where this band is selected for pavement layer thickness within the range of 6–10 cm [32]. In this study, total RAP content equal to 100% means that there was no need to incorporate new virgin aggregates in the mix design, but 2.5% emulsion content is added over the weight of RAP.

## 6. Laboratory Compaction Study 

Nowadays, there is no general agreement concerning what the most suitable laboratory compaction method is, nor is there a full consensus regarding the mix design compaction energy that should be selected for the production and compaction of half-warm mix recycling technology with 100% RAP and emulsified bitumen. The compaction method chosen should be capable of reproducing the benchmark density, air voids, and more consistent mechanical performance properties (e.g., indirect tensile strength, rutting, and stiffness modulus) of the laboratory mixtures compared with those values obtained from the road worksite after pavement construction. 

In this context, due to a lack of knowledge on what air voids design criterion should be adopted for this technology, it was decided to support this research study on the real-scale construction project results found by other researchers. For instance, Harmelink et al. [41] evaluated the in situ air voids content of 22 real scale sections for six years and found that, by applying a mix design compaction effort of 75 gyrations, the required target air voids criterion matches the in-situ air voids in the pavement after three years of service life (Vm = 4.0%). Hence, the air voids criterion sought for the half-warm specimens in the laboratory was targeted to be in the order of 3–4%, in the sense that if the air voids design falls below 3%, it may lead to increased rutting susceptibility [42]. To avoid this type of failure, the target apparent density values, by SSD method, should fall within the range of between 2.311 and 2.335 g/cm^3^, considering a maximum specific gravity $\left(\rho_{m}\right)$, by the volumetric method (EN 12697-5:2010) Procedure A, of 2.407 g/cm^3^, for the 2.5% (o/RAP) emulsion content. Therefore, the abovementioned threshold values are considered as part of the benchmark density range to define the most suitable compaction test method and ensure its satisfactory mechanical performance in the field. 

The HWMRA 100%RAP mix production process in the laboratory consisted of mold heating at 85 °C, recycled RAP material (0/5 and 5/25 mm) at 100 °C, adding 2.5% emulsified bitumen (50/70 dmm) at 65 °C, mixture compaction temperature within the range of 70–80 °C, and a prefixed compaction energy depending on the compaction method to be used. Following the laboratory production/compaction process, an accelerated curing/drying treatment of three days (72 h) at 50 °C, was conducted, as part of the optimization of the mix design process, according to the Spanish technical specifications edition in Art. 20 of PG-4: “In-situ recycling of bituminous mixtures with emulsion” [32]. More details on the curing process can be found in Section 7.2.

### 6.1. Marshall Impactor Hammer

In this first phase, the Marshall specimens were prepared (with a diameter of 101.6 mm and a height of 63.5 ± 1.5 mm) and compacted by both the Marshall impactor (EN 12697-30:2012) and the gyratory compactor (EN 12697-31:2007) in order to determine if there was a possible correlation between the results shown by both compaction methods. To this end, the Marshall specimens were compacted by applying two different compaction energies, i.e., 75 and 100 impact-blows on each side, and 70 gyrations with the gyratory compactor, respectively. Following the compaction process, the apparent density, by SSD conditions, stiffness modulus at 20 °C, and indirect tensile strength at 15 °C, were determined and compared with those results obtained from the gyratory compactor. In this context, the average apparent density value of 75 impact-blows was found to be 2.282 g/cm^3^, and an average air voids content of 5.2%, while, for 100 impact blows, this number was 2.297 g/cm^3^ and 4.6% air voids.

On the other hand, by applying 100 impact-blows, the average stiffness modulus value of the HWMRA 100% RAP mixture was 2,473 MPa, and an average indirect tensile strength value of 1.18 MPa; whereas, for 75 impact-blows, the stiffness modulus and indirect tensile strength values were somewhat similar to those values obtained from the Marshall hammer with 100 impact-blows.

Concerning the gyratory compactor test method, it was observed that, by applying a mix design compaction energy of 70 gyros and following the current laboratory standard test conditions established (0. 82°, 30 rpm, and 600 kPa) by the EN 12697-31:2007 standard, the average stiffness modulus value of 3,134 MPa was obtained, and an average indirect tensile strength value of 2.02 MPa. In other words, the laboratory specimens compacted with the Marshall impactor hammer displayed a significant decrease of the stiffness modulus values of 20%, and lower indirect tensile strength, ITS in-dry, values in the range of 34–42%. Analogous outcomes for recycled mixes compacted with the Marshall impactor were found in other laboratory studies. Hartman et al. [43] claim that the Marshall compactor does not have a kneading effect to re-orientate the particle size distribution, and, hence, produces lower density, increased stiffness, and mechanical properties that differ considerably from the values obtained in the field cores [44,45,46,47]. For this reason, the Marshall impactor was not considered for further testing, since it delivers lower volumetric characteristics (e.g., air voids and bulk density) and mechanical performance properties (i.e., indirect tensile strength and stiffness modulus) than those obtained with the gyratory compactor.

### 6.2. Static Compressive Load by Double-Plunger 

In this research study, the vertical static compressive stress load (NLT-161/00: Standard Test Method for Compressive strength of Bituminous Mixtures) [36] was evaluated for half-warm specimens’ preparation (with a diameter of 101.6 mm and 100 mm in height) and characterization in the laboratory.

In this context, looking at the mixtures’ compressive strength results, it was observed that, by applying a static compressive load of 21 MPa, the HWMRA 100% RAP mixture with 2.5% (o/RAP) emulsion showed an average apparent density, by SSD method, of 2.357 g/cm^3^, and an average air voids content of 2.08%; whilst the average compression strength value was found to be 5.63 MPa. Despite this, it was found that the static compressive stress results were much higher than the minimum threshold values required by the Spanish technical standards, according to Art. 20 of PG-4 [48]. Analogous results for the static compressive test have been found and contrasted in other laboratory studies conducted by Hartman et al. [43] and Martínez et al. [49]. They claim that the compressive system is not regarded as the most reliable laboratory compaction method since it always provides much higher density values than those obtained in the field cores. The increased density is attributed to a higher contact load pressure applied on the cylindrical specimens, causing the breakage of aggregates and squeezing of the binder, resulting in a much higher density than the obtained from the field cores [50]. Moreover, since these specimens are prepared with a height of 100 mm, they cannot be reused for further mechanical testing (i.e., indirect tensile strength and stiffness modulus).

In view of that, and based on the author’s findings and laboratory results, the cylindrical specimens had to be prepared with new pre-fixed dimensions (i.e., with a height of 60 mm and with a diameter of 101.6 ± 0.1 mm) and compacted with a lower static compressive pressure of 10 MPa. In other words, a 53% lower compressive stress energy was selected to obtain specimens with field-like density, and more consistent mechanical properties compared to those values achieved in the road worksite after pavement construction.

In summary, for 10 MPa static load, the average apparent density, by SSD, was found to be 2.289 g/cm^2^, and air voids content of 4.9%, resulting in slightly higher air voids content than those values expected from the target air voids content set in the mixture design. Thus, the average indirect tensile strength (ITS) value was found to be 1.66 MPa and an average stiffness modulus of 3,578 MPa when applying the selected static compaction pressure of 10 MPa. However, at the end of the compaction process, it was observed that this latter compaction energy caused the breakage of aggregates, showing that the static load was not the most suitable compaction method for this technology.

### 6.3. Gyratory Compactor 

The shear gyratory compactor (SGC) was also selected to determine the volumetric and mechanical properties of the mixtures, according to EN 12697-31:2007. Part 31: Gyratory Compactor. The laboratory standard compaction conditions included: (1) an internal angle of 0. 82°; (2) constant speed of 30 rpm; (3) vertical consolidation pressure of 600 kPa; (4) compaction temperature of 80 °C; and (5) number of compaction gyrations: variable. In fact, this compaction test method is typically adopted as the most appropriate laboratory compaction system to achieve the benchmark density, a more even homogeneous air voids distribution [51,52], more consistent engineering properties to those obtained in the field cores [53], and for simulating field compaction conditions because of the effect of kneading motion [43,54,55,56]. For all these reasons, the gyratory compactor system was picked and evaluated as benchmark method for the HWMRA specimens’ production and characterization in the laboratory, since it allows to provide a better simulation of compaction than other compaction methods as well as monitor the specimen’ height change through the densification curve [57,58].

In this regard, an assessment of the effect of two emulsion contents (2.5% and 3.0% o/RAP) and two penetration grade bitumen (160/220 and 50/70 dmm) was conducted to determine the way in which the volumetric characteristics behave depending on the mix design compaction effort used; where the optimum emulsion content (OEC) was defined as the amount required to achieve the target air voids content in the range of 3–4% at a given number of compaction gyrations. Figure 4 shows the thermographic analysis of the specimens together with the thermography of the cylindrical mold heated at 85 °C. To do so, a FLIR C2 thermographic imaging camera (with a normal temperature emissivity of 0.95, surface reflexivity temperature of 20 °C and infrared (IR) resolution of 80 x 60 pixels) and FLIR tool software were used to obtain the main parameters during the production and compaction process (i.e., maximum, minimum, and average temperature), wherein the central rectangular-shaped section of the cylindrical mold was recorded for ensuring the target compaction temperature (~80 °C). However, the actual temperature monitoring of the specimen was conducted using a thermal probe that is inserted into a hole of the mold in order to control that the working temperature was approximately 80 °C during the mix compaction process.

Figure 5 illustrates the specimens’ geometric density (g/cm^3^) change and air voids content (%) curve against compaction gyrations, ranging from 0 to 200 gyrations. In this regard, the compaction curves showed in this figure are determined by means of the geometric density, according to EN 12697-10:2010: Part 10: Compactibility, whilst the bulk density showed in Table 3 is calculated at the end of the compaction process, by saturated surface dry (SSD) conditions (EN 12697-6:2012. Part 6: Determination of bulk density of bituminous specimens). The geometric density is, therefore, calculated with the change of the geometric volume of the specimens, and it relies on the evolution of the specimen’s thickness.

Concerning the slope of densification curve of the 100%RAP mixture with 2.5% (o/RAP) emulsion, it can be observed in Figure 5 that, by following the laboratory standard compaction conditions set in EN 12697-10:2010, there is a significant decrease in the air voids curve when increasing the number of load cycles derived from an initial mix densification of 13.3% (i.e., $\rho_{b,dim}$ increased from 2.040 to 2.311 g/cm^3^) during the first 70 gyrations. Afterward, an aggregate–aggregate interlocking was observed in the compaction range of 70–100 gyrations, which led to an increase in density of approximately 0.61% in 30 gyrations (i.e., from 2.311 to 2.325 g/cm^3^); whereas at the end of the compaction test (200 gyrations), the slope of the densification curve became more stable and asymptotic thereafter, i.e., it reached a slight increase in the average geometric density of 0.77% in the last 100 gyrations applied on the specimen ($\rho_{b,dim}$ = 2.340 g/cm^3^, Vm = 2.5% air voids, at 200 gyrations) In other words, by applying a mix design compaction effort of 70 gyros, the average geometric density was found to be 98.8% of the specimen’ density compacted with 200 gyrations; whereas, for 100 gyrations, this percentage was 99.4%, according to EN 13108-20:2007. Part 20: Type Testing – Annex C. 

In summary, it was found that, for 70 gyrations, the average geometric density of the HWMRA 100% RAP (2.5% o/RAP) mixture with 50/70 pen. bitumen was found to be 2.311 g/cm^3^, an average air voids content of 4.0%; and average indirect tensile strength values of 1.7 MPa. Therefore, the gyratory compaction curve revealed that it is possible to successfully meet the target air voids criterion within the range of 3–4%.

Once the mix design compaction effort was defined, an average of three cylindrical specimens was prepared (with a diameter of 100 mm and 60 ± 1.5 mm in height) and compacted by following the laboratory standard compaction conditions established by the EN 12697-31:2007 standard. Table 3 provides a comparison between the volumetric and mechanical performance value results obtained from the gyratory compactor at 70 gyrations, Marshall impactor hammer at 75 and 100 impact-blows, and a vertical static load of 10 MPa, respectively.

It can be observed in this table that, by applying a mix design compactive effort of 70 gyrations and by setting-up the gyratory compactor with an internal angle of gyration of 0.82° and vertical consolidation pressure of 600 kPa, the highest apparent density, indirect tensile strength (ITSin-dry) and retained water strength values were obtained. This is likely attributed to the kneading effect of the gyratory compactor on the mixtures’ internal structure that allowed the provision of higher apparent density, better rearrangement, and interlocking of the aggregate particles contained in the specimens [59,60,61]—implying that the higher the aggregate–aggregate-interlocking effect, the better the dissipation of the shear stress of the mixture [62].

In this context, one can say that the gyratory compactor was the most suitable compaction test method for the production/compaction and characterization of the half-warm mix recycling technology with total RAP content (100%) and emulsified bitumen. Therefore, Marshall hammer impactor and static compressive strength load by double-plunger were not considered for further testing analysis, since they tend to induce higher mechanical impact stress load and static compressive load on the cylindrical specimens, which results in the breakage of aggregates and, hence, the weakening of the mixture’ mechanical performance.

Additionally, an assessment of the volumetric and mechanical performance properties was conducted to determine the feasibility of using half-warm mix asphalt mixtures with two emulsion contents (2.5% and 3.0%) and two penetration grade bitumen (160/220 and 50/70 dmm), as illustrated in Table 4. In this sense, for the HWMRA mixture with 2.5% emulsion and 50/70 and 160/ 220 pen. grade bitumen, the highest stiffness modulus values of 2988 and 2901 MPa were obtained, respectively; while, for the HWMRA mixture with 3.0% emulsion, this number was much lower than the 2.5% emulsion. Nevertheless, only 2.5% o/RAP emulsion met the required air voids content within the range of 3–4%.

Therefore, it is worth noting that the 2.5% emulsion with 50/70 pen grade bitumen meets the minimum indirect tensile strength ratio (ITSR ≥ 70%) stipulated for their use in low traffic load categories and shoulders, as well as the ITSR ≥ 75% for their use in base and binder course asphalt mixtures and intermediate traffic load categories, according to the latest Spanish technical regulations edition in Art. 20 of PG-4 [32]. Moreover, the HWMRA mixtures meet the minimum indirect tensile strength ratio values required for hot mix asphalt mixtures for their use in base, binder (ITSR ≥ 80%), and wearing course asphalt mixtures (ITSR ≥ 85%) of road pavements, according to Art. 542 of PG-3 [63].

## 7. Mixture Design

### 7.1. Determining Optimum Emulsion Content 

Once the aggregate gradation curve, emulsion-type, and mix design compaction energy were defined, the next step was to proceed with the manufacturing of new cylindrical specimens to calculate the optimum emulsion content (OEC) by testing a wide range of emulsion contents (0%, 2.0%, 2.5%, 3%, and 3.5% o/RAP), depending on the volumetric and mechanical performance properties. The HWMRA mixes were manufactured by heating the RAP at 100 °C, bituminous emulsion at 65 °C, mix fabrication at 95 °C, and mix compaction temperature in the range of 70–80 °C using prefixed compaction energy of 70 gyrations. The optimal mixture design consisted of defining the optimal emulsion content that allowed to meet the laboratory performance requirements regarding apparent density, air voids content in the range of 3–4%, indirect tensile strength (ITS) at 15 °C, stiffness modulus at 20 °C, water sensitivity, and resistance to permanent deformation at 60 °C. To this end, an average of three cylindrical specimens was prepared (with a diameter of 100 mm and 60 ± 1.5 mm in height) and tested in the laboratory by considering variations of the emulsion content to be used in the mix design, as illustrated in Table 5. 

Figure 6a displays the air voids content and apparent density values of the mixtures designed with 50/70 pen. bitumen, while in Figure 6b, the stiffness modulus, at 20 °C, and indirect tensile strength, at 15 °C, versus five different emulsion contents, ranging from 0% to 3.5%, were plotted. 

In this context, it was observed in Figure 6b that the 100% RAP mixture with 0% emulsion content exhibited lower indirect tensile strength (ITS) values (1.08 < 1.5 MPa), increased stiffness modulus, and lower moisture damage (69% < 75%) values than those minimum required by the Spanish technical specifications in Art. 20 of PG-4 [32]. The decreased water susceptibility values can be partially attributed to the loss of the adhesive bonding between aggregates and binder, i.e., due to the failure of the cohesive strength of the binder—although it is expected that such a stripping effect can be counteracted when adding the respective emulsion content to be used in the preliminary mix design [12,64]. As for the stiffness modulus values, the HWMRA 100% RAP mixture with 2.5%o/RAP showed a decrease in the stiffness of 23%, in comparison with the 0% emulsion. This result implies a positive aspect to improve the mixtures’ fatigue cracking resistance in the field since it would make the mixture less stiff and less brittle by enabling higher tensile deformations before its fatigue cracking failure occurs in the field. 

Therefore, based on Table 5 and Figure 6, one can say that the 100% RAP mixture with 2.5%o/RAP emulsion was found to meet the target air voids content (Vm = 3.3%), internal cohesion (ITSdry) above 2.0 MPa and moisture sensitivity values above 75%, according to Spanish specifications in Art. 20 of PG-4 [32], as well as the stringent moisture damage requirements (ITSR>85%) for hot mix asphalt (HMA) mixtures in wearing course asphalt mixtures of road pavements, according to Art. 542 of PG-3 [63].

### 7.2. Influence of the Curing Process on the Mechanical Performance

Once the optimal emulsion content (2.5% o/RAP) was selected, the next step was to quantify the way in which the accelerated curing process promotes the development of the ultimate mechanical performance properties (e.g., ITS and stiffness modulus) of the mixtures. In this sense, the curing treatment was conducted, as part of the optimization process of the mix design, using a forced-draft convection oven, at 50 °C, at four different curing periods (0, 24, 48, and 72 h), at 24 h increments, until reaching constant weight before their testing according to Art. 20 of PG-4: “In-situ recycling of bituminous layers with bitumen emulsion” [32]. Following the laboratory curing process, the specimens were removed from the molds to verify if there was residual moisture loss, either because of evaporation or chemical reactions caused by entering into contact with the aggregates on compacted specimens [65]. Thus, most of the residual water content of the emulsion is lost during the compaction and curing process because of water migration and moisture loss by evaporation, in which the half-warm emulsified specimens were found to reach a constant weight at 72 h and 50 °C.

The load-bearing capacity of the mixtures was evaluated through the stiffness modulus test, at 20 °C, according to EN 12697-26:2018. Part 26: Stiffness. As it can be observed in Table 6 and Figure 7, the stiffness modulus values exhibited a steeper upward curve as a result of the increase in curing time, which led to a higher maximum peak stiffness modulus value of 3462 MPa—showing a significant improvement in the average load-bearing capacity of 13% (i.e., from 24 to 72 h). The likely explanation is attributed to a higher residual internal friction between the aggregate particles because of moisture loss by evaporation and breakage of the emulsion [66]. Therefore, it can be said that the use of a curing treatment is assumed as a positive aspect to improve the indirect tensile strength, stiffness modulus, and rutting performance [67]. Analogous results for mixes in need of curing process were found in other laboratory studies. Kim and Lee. [68] claimed that mixtures’ mechanical performance (i.e., flow number, ITS, and dynamic modulus) improves as a consequence of the increase in curing time [69,70]; whilst some other authors reported that mixtures containing emulsified bitumen and fabricated at low temperatures tend to require an accelerated curing treatment to further develop the ultimate mechanical performance properties (e.g., stiffness modulus and ITS) at the early hours of being placed in the field [66,71,72]. In this recognition, the increase in the stiffness modulus values of the HWMRA 100% RAP (2.5%o/RAP) mixture confirms that these mixtures require a curing process to develop higher internal cohesion and stiffness modulus values.

Figure 7 depicts the indirect tensile strength values of the HWMRA 100% RAP (2.5%o/RAP) mixture with a 50/70 pen. bitumen against the effect of short- (0–24 h) and long-term curing treatment (48–72 h) on the mix performance. As it can be observed in this Figure, the indirect tensile strength (ITS) values remained relatively constant at the early stage of the accelerated curing process. However, at the end of the accelerated curing treatment of three days (72 h), this mixture increased its internal cohesion value by approximately 8%, i.e., the ITS values increased from 2.17 to 2.34 MPa (from 48 to 72 h). The likely explanation for this result lies in the adequate combination of the cohesive strength of the binder and adhesive interface bonding between RAP aggregates and binder, as well as a slight reduction of the air voids content on the plane where the break takes place [73]—implying a higher surface to spread the strain on the specimens. Thus, the use of accelerated curing treatment of three days (72 h), at 50 °C, in a forced-draft convection oven is highly recommended as part of the HWMRA mix production process.

## 8. Advanced Mechanical Characterization of the Mixture

Once the optimum mixture design was defined, the next step was to characterize and compare the mixtures’ mechanical performance results in terms of resistance to permanent deformation (EN 12697-22:2012) and four-point fatigue bending (4PB) beam test method, according to EN 12697-24:2012: Resistance to fatigue—Annex D.

### 8.1. Rutting Performance Results

The characterization to permanent deformation was carried out, as part of the optimization process of the mixture, by conducting the wheel tracking test at two different testing temperatures (50 °C and 60 °C) using the optimal emulsion content (2.5%o/RAP) with a 50/70 pen. grade emulsified bitumen. The wheel tracking test was also conducted, at 50 °C, to simulate and reproduce the real thermal gradients experienced by the binder course asphalt mixture in the field. Table 7 collates the results obtained from the wheel tracking test of the HWMRA 100% RAP mixtures (2.5% o/RAP), which consisted in determining the wheel tracking slope (WTSAIR), calculated between 5000 and 10,000 load cycles, mean rut depth (mm, RDAIR) and the proportional rut depth at 10,000 load cycles (%, PRDAIR).

Though there are no threshold rutting performance values established for this recent technology (Art. 20 of PG-4: In-situ recycling of bituminous mixtures with emulsion) [32], the HMA technical specifications were considered, as part of benchmark rutting values, to evaluate and guarantee its satisfactory performance for binder and wearing course asphalt mixtures of road pavements. 

In this context, the rutting performance values to be met for the purposes listed above are the wheel tracking slope (WTSAIR) value lower than 0.15 (mm/1000 load cycles), between 5000 and 10,000 load cycles, and proportional rut depth value below 5%, according to Art. 542.5.1.3: Resistance to permanent deformation of hot mix asphalt mixtures. In this regard, the wheel tracking slope of the 100% RAP mixture (2.5% o/RAP), at 60 °C, was found to be 0.109 (mm/10^3^ load cycles) and a proportional rut depth of 3.47%. Therefore, this mixture meets the maximum threshold values stipulated for hot mix asphalt mixtures in base, binder, and wearing course asphalt mixtures of road pavements subject to intermediate and low traffic load categories and moderate thermal weather zones in Spain [63]. In this sense, it is highly recommended to include these rutting performance requirements for the drafting of new technical guidelines of this recent technology.

Figure 8 shows that the slope of the 100% RAP mixture curve (2.5% o/RAP), at 60 °C, was slightly steeper by rising sharply during the first 5,000 loading cycles and becomes more stable when applying 8,000 loading cycles with the wheel tracker. Nonetheless, the addition of high RAP contents in asphalt mixtures typically tends to improve the resistance to permanent deformation as a result of the physical hardening and chemical aging (i.e., because of the evaporation of the lighter oil fractions in the bitumen) suffered by the asphalt binder during its service life. Analogous results for mixes containing high and total rates of RAP and additives have been found and contrasted by multiple authors [74,75,76,77,78,79]; whilst the results reported by other authors are rather less conclusive, in the sense that mixtures manufactured at low temperatures and emulsified bitumen are typically characterized by having a lower rutting performance than that of the conventional HMA mixtures [15]. 

On the other hand, it was observed that the 100%RAP mixture (2.5%o/RAP), at 50 °C, exhibited an average wheel tracking slope value of 0.068 (mm/10^3^ load cycles), between 5000 and 10,000 load cycles, and an average rut depth value of 1.42%. This result implies that the wheel tracking test results, in general, depend more on the test temperature than on the recycled content added, even for the HWMRA 100%RAP mixture.

### 8.2. Fatigue Cracking Test (4PB)

To complete the characterization of the mechanical performance of the mixtures studied, the four-point fatigue bending beam (4PB) test method was conducted, at 20 °C, using a loading frequency of 30 Hz, according to EN 12697-24—Annex D. In Figure 9, the mixtures’ fatigue cracking resistance laws for each type of pen. grade bitumen (50/70 and 160/220 dmm) were evaluated. The determination coefficients, (R^2^), fell in the range of 90-95%, indicating a good level of correlation. Thus, it can be said that, for the strain–fatigue levels tested, the HWMRA mixes with both 50/70 and 160/220 pen. bitumen showed comparable fatigue slopes, suggesting that they have an equivalent sensitivity to strain in terms of fatigue life. Nevertheless, the fatigue life of the HWMRA 100% RAP with 160/220 pen. grade bitumen was found to be slightly better than that of the 50/70 bitumen, likely attributed to the effect of a softer penetration grade bitumen in the final mixture design that promoted the extension of the mixture’s fatigue life. 

In this context, for the HWMRA 100% RAP (2.5%, o/RAP) mixtures with 50/70 pen. emulsified bitumen, the average flexural modulus was found to be 6331 MPa (with a standard deviation (SD) of 465 MPa and a coefficient of variation (CV) equal to 7.5%), and strain–fatigue level, ($\varepsilon_{6}$), at 10^6^ load cycles, of 143 µm/m; whereas, for the 160/220 pen. grade emulsified bitumen, the average flexural modulus of 5936 MPa was obtained, and an average micro-strain fatigue level of 155 µm/m. Analogous results for 4PB fatigue cracking and fracture properties have been found in several laboratory fatigue studies conducted by different researchers [24,25,80] for a WMA/ HMWA mixture with a total RAP content (100%) and emulsified bitumen. 

## 9. Conclusions

The primary objective of this research study was to evaluate three different laboratory compaction test procedures (i.e., Marshall impactor, static load by double-plunger and gyratory compactor) and define the most suitable method for half-warm mix recycled asphalt mixtures containing a RAP content equal to 100% and emulsified bitumen. The second aim was to quantify the effect of accelerated curing treatment on the development of mechanical performance properties of the mixtures. For the HWMRA specimens’ compaction and characterization in the mix design stage, the main findings and results that can be drawn from this research phase are summarized below:Looking at the volumetric and mechanical performance properties obtained from three different laboratory compaction methods, one can say that the gyratory compactor system turned out to be the most suitable compaction test method for half-warm mix recycled asphalt (HWMRA) specimens’ production/compaction and characterization in the laboratory. To do this, the specimens were compacted by applying a mix design compaction energy of 70 gyros, at 80 °C, and setting up the gyratory compactor at an internal angle of gyration of 0.82°, vertical consolidation pressure of 600 kPa, and speed of gyration of 30 rpm.It is worth noting that the use of static compressive stress load by double-plunger of 21 MPa load was discarded for further mechanical testing since this method exhibited a much higher bulk density, indirect tensile strength, and stiffness modulus values than those obtained from the road worksite after pavement construction. Moreover, it was observed that the static method caused the breakage of aggregates and binders’ squeezing throughout the mix compaction process.Concerning the Marshall impactor results with 75 and 100 blows on each side, a significant worsening in the volumetric (air voids and bulk density) and mechanical performance (stiffness modulus and ITS) of the mixtures was obtained, in comparison with the results obtained from the gyratory compactor at 70 gyrations; likely as a result of the breakage of aggregates during the mix compaction process.The effect of a long-term accelerated curing treatment of three days (72 h), at 50 °C, was found to be rather positive on the development of the mixtures’ mechanical performance properties in terms of stiffness modulus and indirect tensile strength.The HWMRA 100% RAP with 2.5%o/RAP mixture meets the moisture damage and rutting performance values established by the Spanish technical specifications for recycled mixtures with emulsion for their use either in intermediate or low traffic load categories of road pavements. Furthermore, this mixture was found to meet the requirement values stipulated for hot mix asphalt mixtures for these types of layers (base course asphalt mixtures) and thermal weather.For the same strain levels tested, one can say that the HWMRA mixes with 50/70 pen. grade bitumen showed acceptable performance in terms of fatigue life. However, 50/70 pen. bitumen exhibited slightly lower microtensile fatigue–strain ($\varepsilon_{6}$) values than the results from the 160/220 pen. grade bitumen. This is likely attributed to the effect of a softer penetration grade bitumen in the final mixture design that allowed the provision of higher ductility and flexibility of the mixture by enabling greater tensile-strain fatigue loads.

As for upcoming research lines, it is intended to keep working on the monitoring and characterization of the mechanical performance of HWMRA mixtures with 100% RAP and emulsified bitumen, either in-laboratory or in-field, to encourage higher confidence in promoting the use of these mixtures for maintenance and rehabilitation (M&R) activities of road pavements.

## Figures and Tables

**Figure 1 materials-12-01992-f001:**
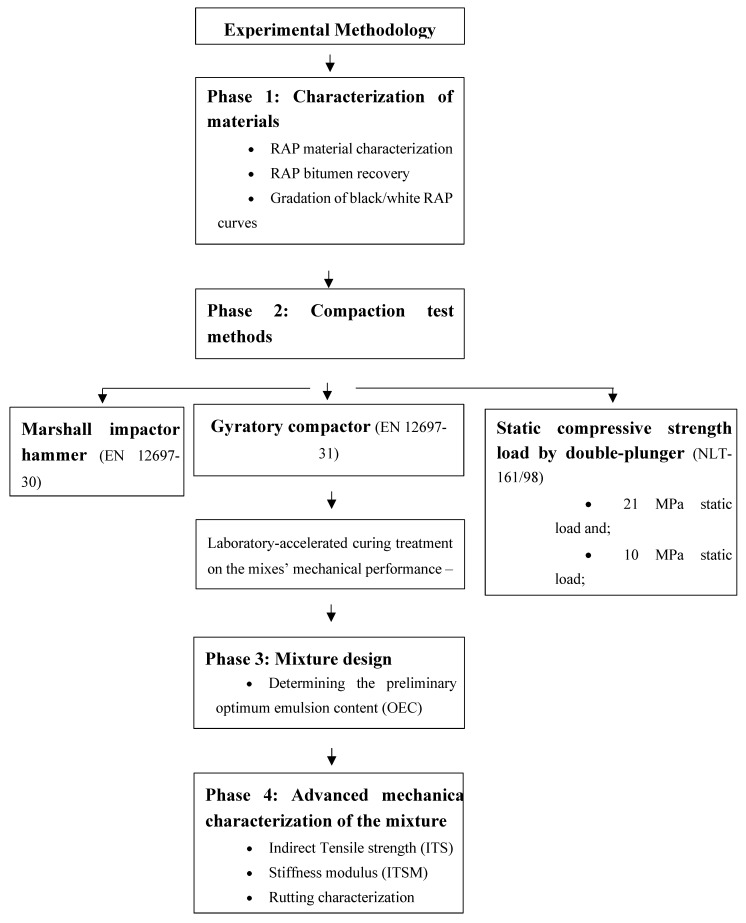
Detailed flow chart of the experimental methodology of the research study.

**Figure 2 materials-12-01992-f002:**
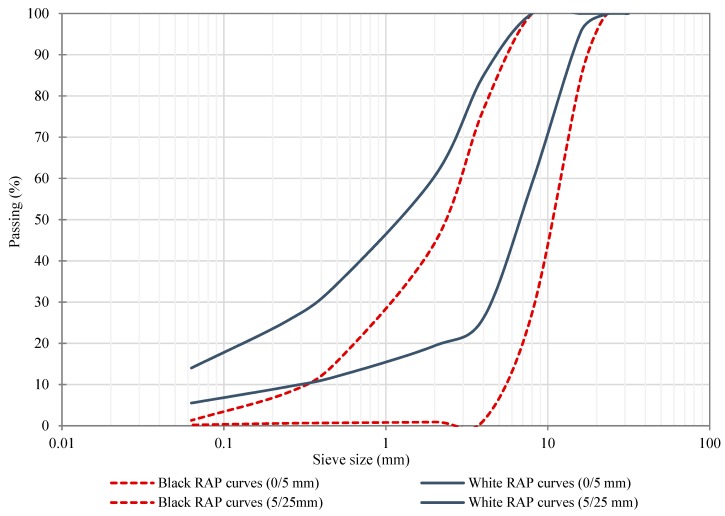
White and black grading curves for both RAP fractions.

**Figure 3 materials-12-01992-f003:**
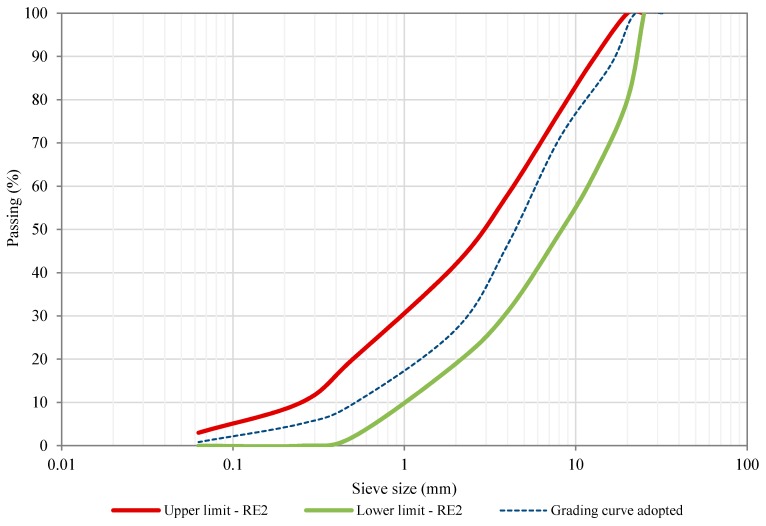
RE2 aggregate gradation sieve sizes and aggregate grading curve adopted.

**Figure 4 materials-12-01992-f004:**
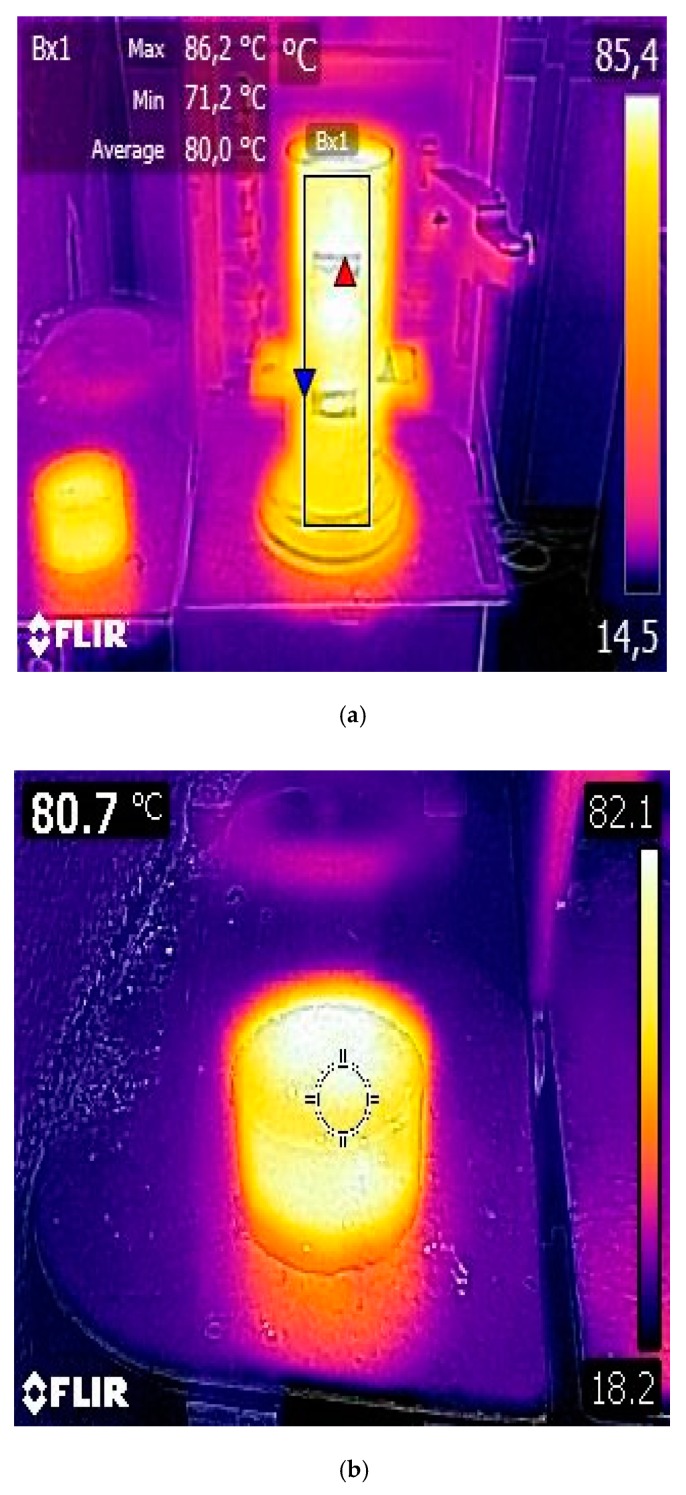
Thermographic imaging analysis: (**a**) mold heating temperature within the range of 80–85 °C; and (**b**) half-warm specimen compacted at 80 °C.

**Figure 5 materials-12-01992-f005:**
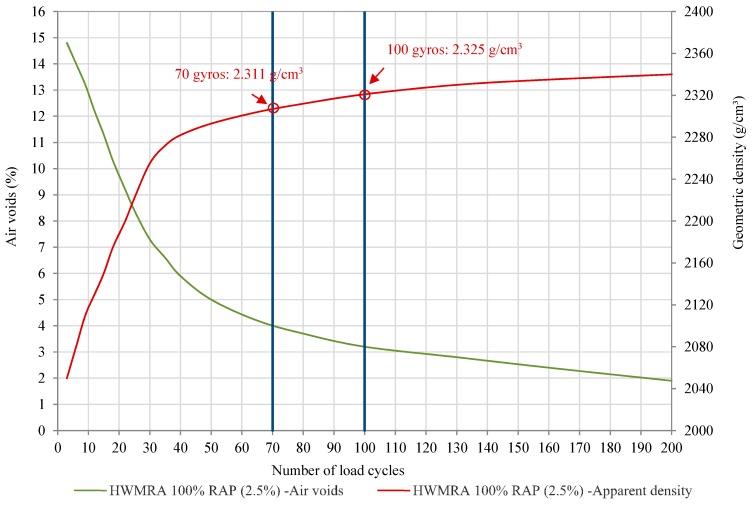
Volumetric characteristic results of the HWMRA 100% RAP mixture with 2.5% emulsion content.

**Figure 6 materials-12-01992-f006:**
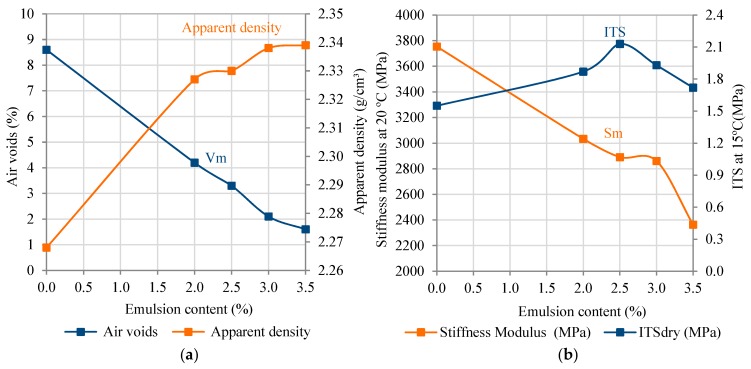
Volumetric and mechanical performance results of the HWMRA 100% RAP mixtures: (**a**) air voids content and apparent density versus emulsion content; and (**b**) stiffness modulus at 20 °C, and ITS, at 15 °C, versus emulsion content.

**Figure 7 materials-12-01992-f007:**
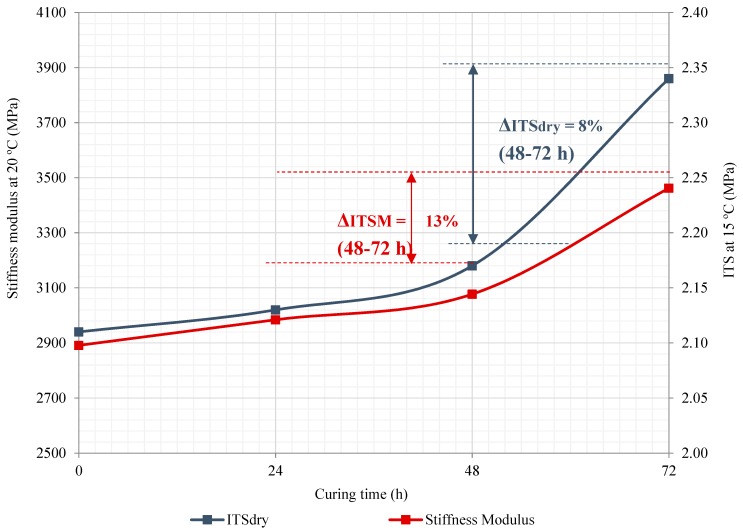
Effect of short- and long-term curing treatment on the stiffness modulus, at 20 °C, and ITS, in dry, at 15 °C versus curing time (h).

**Figure 8 materials-12-01992-f008:**
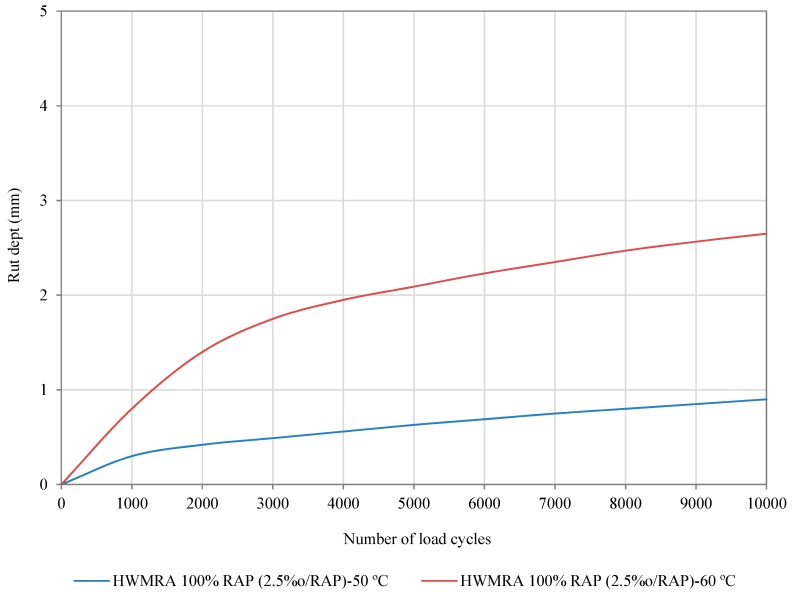
Wheel tracking test results of the HWMRA 100%RAP mixtures with 2.5% emulsion and 50/70 pen. bitumen.

**Figure 9 materials-12-01992-f009:**
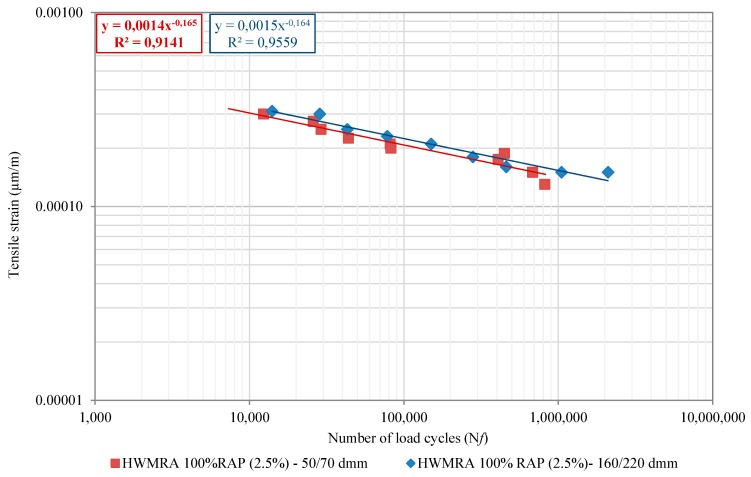
Fatigue cracking resistance laws of the HWMRA 100% RAP mixtures with 50/70 and 160/220 pen. bitumen.

**Table 1 materials-12-01992-t001:** Technical characteristics of the cationic bituminous emulsions (C60B5).

Characteristics	Test Method	Unit	C60B5 160/220	C60B5 50/70
Penetration, at 25 °C (100 g, 5 s)	EN 1426	0.1 dmm	183	66
Residual bitumen content (from water content)	EN 1428	%	61	61.2
Water content	NLT 137	%	39	38.8
Recovered oil distillate from emulsion by distillation	EN 1431	%	0	0
Saybolt-Furol Viscosity, at 25 °C	EN 12846-1	s	23	26
Storage stability by sieving (0.5 mm sieve size)	EN 1429	%	0.01	0.01
pH	NLT 195		3.0	3.0

**Table 2 materials-12-01992-t002:** RE2 aggregate gradation band [32] and gradation curve adopted (%, passing).

Sieve Size (mm)	32	22	16	8	4	2	0.50	0.25	0.063
Upper limit—RE2	100	100	89	77	58	42	20	10	3
Lower limit—RE2	100	80	62	49	31	19	2	0	0
Grading curve adopted	100	99.8	88.1	70.96	46.50	26.94	9.64	5.10	0.86

**Table 3 materials-12-01992-t003:** Comparison between volumetric and mechanical properties of the 100% RAP mixes with 2.5% emulsion and 50/70 pen. bitumen.

Properties	Test Method	Marshall Impactor *		Static Stress Load *	Gyratory Compactor
Compaction energy	-	75 × 2	100 × 2	10 MPa	70 gyros
Specimens’ height, (mm)	-	-	-	67.3	65.2
Apparent density, SSD, (g/cm^3^)	EN 12697-6:2012	2.282	2.297	2.289	2.331
Air voids, (%)	EN 12697-8:2003	5.2	4.6	4.9	3.2
ITS, in-dry, (MPa)	EN 12697-23:2018	1.18	1.33	1.66	2.02
ITSR, (%)	EN 12697-12:2012	-	-	79	95.7

* **Key:** Marshall impactor and static stress load methods were not considered for further mechanical testing since they caused the breaking of aggregates during the mix compaction process.

**Table 4 materials-12-01992-t004:** Volumetric and mechanical performance properties of the mixtures compacted with 70 gyrations.

Mixture Properties	Test Method	Hwmra 100% RAP Mixture			
		Rejuvenator Binder (160/220 dmm)		Residual Binder (50/70 dmm)	
Emulsion (%, o/RAP)	-	**2.5%**	3.0%	2.5%	3.0%
Height, (mm)	-	65.0	64.9	65.5	65.4
Apparent density, SSD, (g/cm^3^)	EN 12697-6	2.347	2.350	2.340	2.344
Air voids, Vm, (%)	EN 12697-8	2.98	2.47	3.08	2.49
ITSdry, 15 °C, (MPa)	EN 12697-23	2.14	2.06	1.99	1.67
ITSwet, 15 °C, (MPa)	EN 12697-23	2.05	1.91	1.91	1.57
ITSR, (%)	EN 12697-12	95.7	92.7	95.8	94
Stiffness modulus, 20 °C, (MPa)	EN 12697-26	2901	2389	2988	2560

**Table 5 materials-12-01992-t005:** Volumetric and mechanical performance results of the mixtures.

Mixture Properties	Test Method	Emulsion Content (%, o/RAP)—50/70 dmm				
		0%	2.0%	2.5%	3.0%	3.5%
Maximum density (g/cm^3^)	EN 12697-5	2.481	2.428	2.407	2.389	2.377
Apparent density, ssd, (g/cm^3^)	EN 12697-6	2.268	2.327	2.328	2.338	2.339
Air voids, Vm, (%)	EN 12697-8	8.6	4.2	3.4	2.1	1.6
ITSdry, (MPa)	EN 12697-23	1.55	1.87	2.13	1.93	1.72
ITSwet, (MPa)	EN 12697-23	1.08	1.82	2.08	1.89	1.69
ITSR (%)	EN 12697-12	69.5	97.3	97.6	98.1	98.3
Stiffness modulus, 20 °C, (MPa)	EN 12697-26	3754	3034	2891	2861	2364

**Table 6 materials-12-01992-t006:** Mechanical and volumetric property results of the mixtures after curing/drying treatment.

Mixture Properties	Test Method	Curing Time (h)—2.5% Emulsion				
		0	24	48	72	ΔITS/ITSM 0–72 h
ITS, in-dry, 15 °C, (MPa)	EN 12697-23:2007	2.11	2.13	2.17	2.34	11%
Stiffness modulus, 20 °C, (MPa)	EN 12697-26:2007	2891	2984	3077	3462	20%

**Table 7 materials-12-01992-t007:** Wheel tracking test results of the mixtures.

Mixture Properties	Test Method	HWMRA 100% RAP	
		(2.5%)—50 °C	(2.5%)—60 °C
Apparent density, by SSD, (g/cm^3^)	EN 12697-6	2.302	2.328
Deformation at 5.000 load/cycles, RDAIR, (mm)	EN 12697-22	0.52	2.11
Deformation at 10.000 load/cycles, RDAIR, (mm)	EN 12697-22	0.86	2.66
Wheel tracking slope, WTSAIR, (mm/10^3^ load cycles)	EN 12697-22	0.068	0.109
Proportional rut depth, PRDAIR, (%)	EN 12697-22	1.42	3.47

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
