# Peer review of "Laboratory Compaction Study and Mechanical Performance Assessment of Half-Warm Mix Recycled Asphalt Mixtures Containing 100% RAP"

_materials, 2019, doi:10.3390/ma12121992_

Round 1
Reviewer 1 Report
This paper cannot be accepted in such a form because of the following concerns:
1. In this paper, half warm mix asphalt was used. Please define cold mix asphalt, half warm mix asphalt, warm mix asphalt, and hot mix asphalt, respectively. Related references are also required.
2. Recently, a larger number of works have been done aiming at increasing the percentage of RAP to 100%. While, there still remain questions in terms of the definition of 100 % RAP. Does the 100% RAP refer to asphalt mixture containing only RAP without any newly-added aggregate or asphalt binder? If not, what does the 100 % RAP really mean? Please provide references. The reviewer thinks clarify this kind of basic concept is of critical importance.
3. Line 791. Same stress levels? or it should be same strain levels? Please double check it.
4. Strain-control model was selected in the 4PB test. What about the fatigue behavior of HWRMA at stress-control model?
5. Line 478. The IR image was provided. However, IR is more reflective of the temperature of the surface of the test object. Is it still meaningful to measure the temperature of asphalt mixture within a metal model using IR camera?
6. What about the dynamic modulus of HWRMA?
7. To support the conclusion, comparison between SHAWARMA and HAM should be performed.
This paper cannot be accepted in such a form because of the following concerns:
1. In this paper, half warm mix asphalt was used. Please define cold mix asphalt, half warm mix asphalt, warm mix asphalt, and hot mix asphalt, respectively. Related references are also required.
2. Recently, a larger number of works have been done aiming at increasing the percentage of RAP to 100%. While, there still remain questions in terms of the definition of 100 % RAP. Does the 100% RAP refer to asphalt mixture containing only RAP without any newly-added aggregate or asphalt binder? If not, what does the 100 % RAP really mean? Please provide references. The reviewer thinks clarify this kind of basic concept is of critical importance.
3. Line 791. Same stress levels? or it should be same strain levels? Please double check it.
4. Strain-control model was selected in the 4PB test. What about the fatigue behavior of HWRMA at stress-control model?
5. Line 478. The IR image was provided. However, IR is more reflective of the temperature of the surface of the test object. Is it still meaningful to measure the temperature of asphalt mixture within a metal model using IR camera?
6. What about the dynamic modulus of HWRMA?
7. To support the conclusion of this study, comparison between SHAWARMA and HAM should be performed.
Author Response
To: Editors of Materials
We are more than grateful for the useful feedback by each of the Reviewers that helped us to further enhance and strengthen the quality of the submitted research manuscript. For these reasons, we acknowledge the time and effort that the Reviewers have put into the submitted edition. Thus, we have carefully answered to all points, questions, concerns, and, thereafter, modified the manuscript accordingly.
Response to reviewer’ comments 1:
This paper cannot be accepted in such a form because of the following concerns:
1. In this paper, half warm mix asphalt was used. Please define cold mix asphalt, half warm mix asphalt, warm mix asphalt, and hot mix asphalt, respectively. Related references are also required.
DONE: The following paragraph has been added in the revised manuscript
whereas conventional hot mix asphalt (HMA) are manufactured at 160 ºC, warm mix asphalt (WMA) fabricated in the range of 110-140 ºC, and cold mix asphalt (CMA) below 60 ºC [2].
Page 2. Line 50-52
2. Recently, a larger number of works have been done aiming at increasing the percentage of RAP to 100%. While, there still remain questions in terms of the definition of 100 % RAP. Does the 100% RAP refer to asphalt mixture containing only RAP without any newly-added aggregate or asphalt binder? If not, what does the 100 % RAP really mean? Please provide references. The reviewer thinks clarify this kind of basic concept is of critical importance.
DONE: The following paragraph has been added in the revised manuscript:
In this study, 100% RAP means that there was no need to incorporate virgin aggregates while the emulsion was added in the new mix design over the weight of RAP.
Page 8. Line 316-317
3. Line 791. Same stress levels? or it should be same strain levels? Please double check it.
Thank you for your observation. The word “Stress” has been replaced by STRAIN
4. Strain-control model was selected in the 4PB test. What about the fatigue behavior of HWRMA at stress-control model?
We thank the Reviewer for bringing this point to our attention.
5. Line 478. The IR image was provided. However, IR is more reflective of the temperature of the surface of the test object. Is it still meaningful to measure the temperature of asphalt mixture within a metal model using IR camera?
Thank you for your observation. All of the samples were compacted by following the same compaction protocol set out in the European technical standards, i.e., the mold together with the loose mix were put into the oven for 2 h at 80 ± 10 ºC, according to EN-12697-31:2012. Part 31: Gyratory Compactor. Section 6.2: Preparation of the specimens. Subsequently, the thermography shot was taken right after removing the mold from the oven. Then, it was observed that the mold temperature dropped slowly because of the thermal inertia, suggesting that the mold’s inner face remains at 80 ºC.
6. What about the dynamic modulus of HWRMA?
We appreciate your suggestion. In Spain, the specifications are developed based on the stiffness modulus as a key parameter, according to EN 12607-26:2017. Part 26: Stiffness.
7. To support the conclusion, comparison between SHAWARMA and HAM should be performed.
We thank the Reviewer for the comment. However, the comparison between HWMRA and HMA mixes was removed from the Conclusion section.
Reviewer 2 Report
The paper titled 'LABORATORY COMPACTION STUDY AND MECHANICAL PERFORMANCE ASSESSMENT OF HALF-WARM MIX RECYCLED ASPHALT MIXTURES CONTAINING 100% RAP' was reviewed and the following observations are made:
1) Suggest to include the calculation about the binder content (% by mixture) of the resulting asphalt mixture and the recycled binder content from RAP.
2) Include a few more details on the half-warm mix technology, if possible.
3) The paper is excellently presented: The introduction and background are adequate, the experimental methodology is very clearly laid out, and the results are methodically presented and commented upon. I'd like to congratulate the authors on doing such a wonderful job.
Thanks.
Author Response
Reviewer 2: We would like to thank the Reviewer 2 due to the positive assessment of our research study.
1) Suggest to include the calculation about the binder content (% by mixture) of the resulting asphalt mixture and the recycled binder content from RAP.
Therefore, for the coarse fraction 5/25 mm (60%) the residual binder content was found to be 3.66 (%, o/RAP), whereas, for the 0/5 mm (40%), this content was 7.02 (%, o/RAP). As a result, the content of the aged binder in the RAP (3.66 * 0.6 + 7.02 * 0.4) was 5.004% over the weight of RAP. Additionally, the residual asphalt binder of the emulsion was 60% - suggesting that if 2.5% of emulsion is added to the RAP material (60% * 2.5 % o/RAP), 1.5% of residual asphalt binder is obtained and added to the RAP binder (5.004%), resulting in a total residual binder content of 6.5% o/RAP.
Page 6. Line 205-210.
2) Include a few more details on the half-warm mix technology, if possible.
A paragraph has been added in the introduction section as follows:
Whereas conventional hot mix asphalt (HMA) mixes are manufactured at 160 ºC, warm mix asphalt (WMA) are fabricated in the range of 100-140 ºC, and cold mix asphalt (CMA) below 60 ºC [2].
3) The paper is excellently presented: The introduction and background are adequate, the experimental methodology is very clearly laid out, and the results are methodically presented and commented upon. I'd like to congratulate the authors on doing such a wonderful job.
Thank you. I simply have no words to express my most sincere gratitude.
Reviewer 3 Report
Although the paper is well written and structured, it is not so easy to read and understandable in some of its part. In fact, in some cases, sentences are excessively long, making the reading non-fluent.
Moreover, some paragraph (Introduction and Methodology) can be more summarized.
The Authors gave a lot of attention to describe the procedures (maybe too excessive in some cases) while in other parts adequate details and explanation were missing (Fatigue cracking test paragraph).

Author Response
Reviewer 3:
We would like to thank the Reviewer 3 for bringing this suggestion to our attention.
Although the paper is well written and structured, it is not so easy to read and understandable in some of its part. In fact, in some cases, sentences are excessively long, making the reading non-fluent.
We really appreciate the observation. In some cases, we have split some paragraphs into several sentences to make it easier to read.
1. Moreover, some paragraph (Introduction and Methodology) can be more summarized.
Thank you for your suggestion. We have summarized some paragraphs of the manuscript aiming at trying not to affect the content of the introduction and Methodology.
2. The Authors gave a lot of attention to describe the procedures (maybe too excessive in some cases) while in other parts adequate details and explanation were missing (Fatigue cracking test paragraph).
We really appreciate your suggestion. Some procedures have been shortened.
Reviewer 4 Report
This paper presents interesting results but needs a thorough revision before being considered for publication. Some sections need to be completely rewritten, like the Introduction, literature review and Discussion.
Introduction: The theoretical, analytical and standard approaches should be discussed.
The novelties have to be outlined. It has to be completely rewritten so that the focus of the work and its innovative content can be really appreciated.
Literature review: The Literature review is now a mere list of information but the authors have to provide their own "unifrying" view and not only citing previous work.
Results and discussion: The paper presents a big amount of results from unusual experiments but without a theoretical and practical approach.
Conclusions: The discussion about technological benefit have to be separated in the article according points of conclusions. The analysis of the results is quite basic and deserves better and deeper processing.
Author Response
Reviewer 4: The authors are more than grateful for the helpful feedback given by the Reviewer 4 that helped us to enhance the quality of the research manuscript.
This paper presents interesting results but needs a thorough revision before being considered for publication. Some sections need to be completely rewritten, like the Introduction, literature review and Discussion.
Introduction: The theoretical, analytical and standard approaches should be discussed. The novelties have to be outlined. It has to be completely rewritten so that the focus of the work and its innovative content can be really appreciated. Literature review: The Literature review is now a mere list of information, but the authors have to provide their own "unifying" view and not only citing previous work.
We thank the Reviewer for bringing this point to our attention.
In this sense, we have realized there is no general agreement, nor completely consensus by the pavement community in general, on the mechanical performance and fatigue cracking resistance of mixes manufactured at low temperatures and with total RAP contents, whereby they are still shrouded in uncertainty. Moreover, the origin of these discrepancies is not always easy to determine, because it is about different Materials and test methods. For all these reasos. Spanish materials were selected and evaluated by following the European technical specifications (UNE), so that the potential mechanical performance differences between the current state-of-the-art reviewed and this research manuscript were contrasted and determined.
Additionally, this statement has been added in the revised manuscript.
Therefore, the main contributions of this research paper were the adoption of the gyratory compactor system as the most suitable compaction method for this recent technology; and that the use of an accelerated curing treatment allowed to enhance the mixtures’ mechanical performance in the curing range of between 48 and 72 h, at 50 ºC.
Page 3. Line 117-119
Results and discussion: The paper presents a big amount of results from unusual experiments but without a theoretical and practical approach. Conclusions: The discussion about technological benefit have to be separated in the article according to points of conclusions. The analysis of the results is quite basic and deserves better and more in-depth processing.
We really appreciate the observation. The author's conclusions and findings have been summarized in this manuscript accordingly.
The authors have tried to provide readers with a much better explanation about trends and results obtained in the laboratory. Nevertheless, concerning materials as complex as half-warm mix asphalt (HWMA) mixes, it is not easy to give a detailed explanation of the mixtures’ mechanical performance results, and we want to apologize in this regard. Nevertheless, as soon as this technology spreads and starts gaining momentum, this investigation will contribute to providing further details on the durability and long-term mechanical performance, which makes it possible to clarify some uncertainties regarding this recent technology.
Round 2
Reviewer 1 Report
There are still some concerns need to be kindly addressed.
Considering IR camera is considered to be an tool for measuring the temperature of the outside surface of the test object, why does the author think that a IR camera is a suitable method for measuring the temperature of an asphalt mixture in a metal model?
The reviewer is curious to know the dynamic modulus of the samples.
Author Response
Dear Reviewer,
We really appreciate the useful feedback given by this Reviewer.
Kind regards
The Authors
